# The Impacts of Plastic Waste from Personal Protective Equipment Used during the COVID-19 Pandemic

**DOI:** 10.3390/polym15153151

**Published:** 2023-07-25

**Authors:** Anelise Leal Vieira Cubas, Elisa Helena Siegel Moecke, Ana Paula Provin, Ana Regina Aguiar Dutra, Marina Medeiros Machado, Isabel C. Gouveia

**Affiliations:** 1Environmental Science Master’s Program, University of Southern Santa Catarina (Unisul), Avenida Pedra Branca, 25, Palhoça 88137-270, Brazil; smoecke@gmail.com (E.H.S.M.); ana_provin@yahoo.com.br (A.P.P.); aradutra@gmail.com (A.R.A.D.); 2Environmental Engineering, Federal University of Ouro Preto (UFOP), Ouro Preto 35402-163, Brazil; marina.machado@hotmail.com; 3FibEnTech R&D—Fiber Materials and Environmental Technologies, University of Beira Interior, Rua Marquês d’Ávila e Bolama, 6201-001 Covilhã, Portugal; igouveia@ubi.pt

**Keywords:** COVID-19, Personal Protective Equipment (PPE), plastics wastes

## Abstract

The period from 2019 to 2022 has been defined by the COVID-19 pandemic, resulting in an unprecedented demand for and use of Personal Protective Equipment (PPE). However, the disposal of PPE without considering its environmental impact and proper waste management practices has become a growing concern. The increased demand for PPE during the pandemic and associated waste management practices have been analyzed. Additionally, the discussion around treating these residues and exploring more environmentally friendly alternatives, such as biodegradable or reusable PPE, is crucial. The extensive use of predominantly non-degradable plastics in PPE has led to their accumulation in landfills, with potential consequences for marine environments through the formation of microplastics. Therefore, this article seeks to establish a connection between these issues and the Sustainable Development Goals, emphasizing the importance of efficient management aligned with sustainable development objectives to address these emerging challenges and ensure a more sustainable future.

## 1. Introduction

The years 2019 to 2022 will be recorded in history due to the significant loss of human life that occurred during the COVID-19 pandemic, which was caused by the SARS-CoV-2 virus. The tumultuous circumstances brought about by COVID-19 have not only impacted the healthcare system but have also exerted influences on economic, political, and environmental systems, prompting nations to adopt suppression and mitigation strategies in order to curb transmission within their populations. These measures encompass compulsory social distancing, limitations on non-essential medical services, the closure of non-essential establishments, and the utilization of Personal Protective Equipment (PPE) [1,2].

Nations with resilient healthcare systems and robust economies experienced swift overburdening during the pandemic, prompting attention to shift toward more vulnerable regions of the world, specifically low- and middle-income countries (LMICs) [3]. In these areas, a substantial portion of the population, particularly in impoverished communities such as slums and low-income countries, relies heavily on already strained healthcare systems with inadequate staff and limited resources. Implementing preventive measures like social distancing, regular hand sanitization, and proper waste disposal becomes challenging under such circumstances [4].

Despite the intense efforts of the scientific community in the production of vaccines like Coronavac, which is of Chinese origin, AstraZeneca, which is produced by the University of Oxford, and BioNTech, which is produced by Pfizer, precautionary measures in relation to human contact must remain to reduce the risk of transmitting the SARS-CoV virus-2 [5], including its new variants. Therefore, since COVID-19 was declared a public health emergency on an international scale by the World Health Organization (WHO), several recommendations have been established, including the use of PPE, such as masks and gloves, by health professionals and the rest of the population [6].

Governments have advised their populations to use fabric masks made at home, which can be washed after each use. This is consistent with the objective of reducing the spread of the COVID-19 since the use of a mask hinders the dispersion of droplets and aerosols of the mucous membranes, especially during speaking, coughing, or sneezing. Furthermore, disposable masks are used exclusively by front-line workers to avoid contact with the virus. However, a problem that has arisen is the scarcity of Personal Protective Equipment since it is being constantly replaced to prevent the proliferation of the virus within hospitals and to mitigate the risk of contaminating patients and, importantly, the health professionals themselves [2,6,7].

Consequently, the PPE used by all health professionals and by other citizens has been overloading landfills and the environment, given the amount of waste generated, and the inappropriate disposal of masks and gloves is creating environmental problems [8,9]. The disposal of PPE in nature, as verified by Ocean Asia—ocean conservation in Asia for Asia, results in the direct contamination of ecosystems (in the soil and surface and underground water) with SARC-COV-2, and the death of microorganisms beneficial to the environment can occur indirectly [10]. In addition, the formation of microplastics via weathering is currently an issue of great concern which results from the inappropriate disposal of plastic material. This causes serious problems in terrestrial and aquatic environments, notably in rivers, lakes, and oceans [10,11]. The recent appearance of face masks and gloves as environmental waste is evidence that the global pandemic has contributed to the challenge of reducing plastic pollution in the environment.

In addition to the environmental problems related to the increase in the generation of solid waste, inadequate waste management practices increase the potential for the spread of COVID-19 in developing countries [6]. Therefore, this global emergency has social and economic aspects that extend to environmental issues, such as municipal solid waste (MSW) management, the management of hazardous biomedical waste, and the treatment and disposal of MSW [8,12].

The issue of the escalating generation of waste is among the preoccupations outlined by global leaders and was anticipated within the objectives of the Sustainable Development Goals (SDGs) put forth by the United Nations (UN). These goals are founded upon a collection of widely accepted values, aiming to enhance human living standards, safeguard the planet, and foster prosperity. One of the primary objectives of SDG 12 (specifically, target 12.5) is to significantly diminish waste generation by 2030 through preventive measures, reducing, recycling, and reusing.

Concerning the problem of waste generation in response to COVID-19, the aim of this study was to compile research related to the production of Personal Protective Equipment (PPE) waste during the COVID-19 pandemic. Further, it sought to delve into the feasibility of making PPE either biodegradable or reusable, scrutinizing appropriate strategies for managing such waste and evaluating its prospective impact on attaining the Sustainable Development Goals (SDGs) outlined in the United Nations’ 2030 Agenda.

## 2. Personal Protective Equipment (PPE)

Personal Protective Equipment (PPE), as the name suggests, consists of vital items designed to safeguard lives in the workplace. Within the healthcare field, gloves, masks, white (lab) coats, and glasses play a fundamental role in combating the transmission of diseases, whether through saliva, mucous droplets, aerosols, or other bodily fluids like blood [13,14,15].

N95 masks are recommended as single-use items and are primarily utilized by healthcare professionals treating patients with COVID-19. This is due to their efficient filtration capabilities, as they are able to remove up to 95% of particles with a diameter of 3–5 μm. The mask’s filtration system is composed of electrostatically charged polypropylene layers arranged with microfibers, enabling the effective removal of microorganisms [16].

The medical utilization of N95 masks is relatively recent, having begun in the 1990s as a means to safeguard healthcare workers from drug-resistant microorganisms carried by patients infected with the human immunodeficiency virus (HIV). Subsequently, these masks were employed during the SARS outbreak in 2003. During the current COVID-19 pandemic, these masks are extensively employed by healthcare professionals. Although they are not intended for reuse, the scarcity of such materials during a pandemic is a known challenge. Reusing N95 masks poses the risk of compromising filtration efficiency and a less secure fit on the face, leading to reduced protection [15,16].

Sterilization methods for masks have been studied; the main methods that have been investigated are dry and steam sterilization (autoclaving), the use of vaporized hydrogen peroxide, and ultraviolet germicidal irradiation [17,18]. Some of these approaches are promising but have limitations such as filter wear or alteration, which deteriorate the filtering properties of the mask. Therefore, technologies for sterilization that do not cause major damage to the mask are urgently needed [19].

Another type of mask used is the surgical mask, which has three layers. The innermost layer is in direct contact with the face and absorbs moisture from the user’s breath, and the intermediate layer acts as a filter, while the outer layer repels liquid fluids. Although the outer layer is hydrophobic, dangerous viruses can remain on it, so it is recommended that a surgical mask is used for a few hours and then immediately discarded [16].

Surgical masks are primarily composed of non-woven fabric (NWF) and polypropylene, providing resistance for a maximum duration of 4 h, after which their filtration effectiveness diminishes. Due to the inability to wash or sterilize them, reusing surgical masks is not practical [16]. Although surgical masks are medical items that need to be disposed of within a short time after use, there is a high level of demand for their production.

Regarding the use of disposable gloves, there are no recommendations for the general population to utilize them in routine household tasks. The priority is to allocate their usage to healthcare professionals. Latex, nitrile, and vinyl gloves offer benefits to both the patient and the healthcare professional by preventing direct contact with microorganisms, mucous membranes, blood, and other fluids, regardless of whether they are contaminated or not [20]. The reuse of this type of PPE is expressly not recommended by the World Health Organization since there are still no rapid technologies for effective sterilization [21]. Therefore, the ideal scenario is that the gloves are replaced for each patient, as in the case of surgical masks, resulting in a demand for a high level of production and very fast disposal.

As in the case of masks, it is also important to consider the materials used to produce gloves, since their performance is mainly dependent on the nature of the material used. Gloves with a higher percentage of elongation, for example, are more likely to stretch than tear when pulled, while those with higher tensile strength are more rigid and are more suitable for delicate procedures [22]. In the case of a pandemic, the ideal practice is to use highly resistant gloves, mainly to avoid breakage, perforation, or tears.

The white coats commonly used by health professionals help in avoiding contamination through clothes, serving as shields between the professionals and the patients. This prevents fluids from accidentally reaching the clothes of the health worker, and microorganisms cannot be transferred along the home–work translocation route of contamination. Consequently, it is important to keep these white coats properly cleaned. Industrial washing is an excellent option since this process eliminates any microorganisms [9]. When they are cleaned at home, there is a risk of contaminating the white coat with other non-medical clothing.

In one study, white coats washed at home showed an increase in contamination of 54% at the end of the day, mainly in the regions of the wrists and pockets. Therefore, there is a need to change white coats within short time intervals. In the face of this pandemic, disposable white coats are replaced for each patient, avoiding cross-contamination. In this context, research studies are seeking efficient textile technologies, mainly looking for materials that repel fluids and have antimicrobial agents incorporated into the fabric; however, further discussion and studies regarding the price of these uniforms and their overall health efficacy are needed [9].

Another critical type of PPE utilized in hospital settings, though they are less frequently discussed, are goggles. It is of the utmost importance to emphasize the use of this type of protective equipment as goggles prevent liquids from splashing directly into the eyes and minimize the risk of contamination through touch. It is crucial for the design of goggles to provide excellent peripheral vision while ensuring a secure and comfortable fit [23]. Additionally, some professionals may opt to use a facial protector, but it is essential for the protector to be properly adjusted to the face for optimal effectiveness.

The majority of goggles and face protectors are manufactured using petroleum-based materials like polyethylene. When disposed of, these items generate residues that are challenging to decompose. While the replacement frequency for this type of PPE may not be as high compared to others, there has undeniably been a surge in their utilization due to the ongoing pandemic [23]. As with all PPE, goggles need to be reused safely, and efficient management strategies are required for this waste.

Lastly, it is crucial to emphasize the significance of the proper removal of Personal Protective Equipment (PPE) as healthcare professionals can potentially be exposed to pathogens present on the outer surfaces of equipment. Therefore, comprehensive training should be provided to all healthcare professionals to ensure effective and safe removal of equipment [24]. Additionally, the disposal site should be secure and ideally labeled and sealed to prevent the contamination of third parties, including hospital/clinic cleaning staff and waste collection workers.

Currently, during the COVID-19 pandemic, following government recommendations, a considerable number of individuals have become skilled at producing homemade masks. Consequently, this form of Personal Protective Equipment is now widely utilized on a daily basis across the globe. The effectiveness of this type of mask has been analyzed, and in terms of protection against nanoparticles, it has been demonstrated that masks made with traditional fabrics protect against particles of significantly different sizes. The level of protection varied from 30% to nearly 90%, with some cloth masks offering particle barrier properties similar to commercial surgical masks [25].

Although they guarantee protection against the virus, most masks consist of non-renewable polymers derived from petrochemicals, such as polypropylene, polystyrene, polycarbonate, polyethylene, and polyester, contributing to environmental pollution and subsequent secondary health challenges. In light of the aforementioned discussion, there is an urgent need to quickly develop fully biodegradable facial masks that fulfill the objectives and are low-cost, light, and comfortable [7].

Finally, the importance of conducting studies that contribute to mitigating the adverse effects of the current pandemic, such as the present and future environmental impacts, is highlighted. Additionally, there is a need for research to develop biodegradable face masks derived from natural materials in order to ensure an improved quality of life and the protection of marine and terrestrial ecosystems during this global health crisis [7].

## 3. Personal Protective Equipment and the Generation of Solid Waste during the COVID-19 Pandemic

The COVID-19 pandemic has triggered global emergencies in relation to social and economic aspects which extend to environmental issues, such as solid urban waste management (MSW), the management of hazardous biomedical waste, and the treatment and disposal of MSW [25,26]. Although some positive environmental improvements have occurred due to the lockdown, such as cleaner aquatic ecosystems and reduced air pollution, this is not the case with respect to solid waste management [25,27].

The pandemic has altered the dynamics of waste generation, creating problems for policy makers and workers involved in sanitation [8,27,28]. According to the World Health Organization (WHO), an increase in the volume of infectious waste is expected during the outbreak of COVID 19, and they state that it is necessary to acquire additional treatment capacity by employing alternative technologies such as autoclaves and high-temperature incinerators [29].

Challenges to municipal waste management practices and procedures have arisen, including updating health and safety measures for employees, waste treatment requirements, and general procedures for the waste sector [8,12]. The situation tends to be more critical in developing countries as waste management workers are often not adequately equipped with Personal Protective Equipment (PPE) [8].

Thailand, China, Singapore, and the USA have experienced significant increases in plastic waste generation, encompassing a range of items including face masks, Personal Protective Equipment (PPE), and packaging materials. Notably, Thailand has witnessed a threefold increase in plastic waste production. Additionally, Hubei, China, has observed a concerning surge of 370% in medical waste that is predominantly composed of plastic materials. These statistics emphasize the mounting concern over the increased generation of plastic waste in various regions, underscoring the pressing need for effective waste management strategies and sustainable practices to mitigate the environmental impact caused by such waste [30].

In India, an increase in the generation of yellow biomedical waste (Y-BMW) has been observed during the COVID-19 pandemic. The sudden influx of COVID-infected patients seeking healthcare services has placed a substantial burden on existing incineration units dedicated to the disposal of biomedical waste. On average, each COVID-infected patient in India generated approximately 3.41 kg of biomedical waste per day, with Y-BMW accounting for approximately 50.44% of the total waste generated [31].

Notably, on 13 July 2020, the combined Y-BMW generated by both regular patients and COVID-infected individuals exceeded the incineration capacity of India’s biomedical waste management system. These findings emphasize the urgent need for effective strategies to address the escalating volume of Y-BMW as it poses significant environmental and public health concerns in the country [31].

Singh and Mishra (2021) underscore the significant impact of COVID-19 on India, positioning it as the second-most affected country following the United States. A comprehensive report published on 18 September 2020, shed light on the staggering daily production of biomedical waste in India, which surpassed 180 tons. Notably, the state of Maharashtra emerged as a major contributor, accounting for approximately 17% of the total biomedical waste generated nationwide [32].

The period from June to September 2020 witnessed a substantial surge in the volume of biomedical waste generated in India solely due to the COVID-19 pandemic. In June, the country generated an estimated 3025 tonnes of biomedical waste, followed by approximately 4253 tonnes in July. August recorded an even higher volume, with an approximate generation of 5238 tonnes, while September marked a further increase, reaching around 5490 tonnes. These statistics highlight the unprecedented scale of biomedical waste generation during the specified timeframe, emphasizing the pressing need for robust waste management strategies and infrastructure to mitigate potential environmental and public health hazards [32].

During the COVID-19 outbreak in China, there was a significant 30% reduction in municipal solid waste (MSW) in large- and medium-sized cities due to lockdown measures. However, in Hubei Province, the epicenter of the outbreak, there was a concerning 370% increase in medical waste generation, including infectious and non-infectious waste. This highlights the challenges faced by healthcare facilities and the importance of adapting waste management strategies to handle the surge in medical waste effectively [13].

China experienced a significant surge in the demand for Personal Protective Equipment (PPE), particularly masks, during a specific period. Mask production increased by 450% within one month to meet the heightened demand due to the COVID-19 pandemic. Additionally, the demand for N95 respirators rose from 200,000 to 1.6 million units, underscoring the crucial need for effective respiratory protection among healthcare workers and the general population [27].

These statistics highlight the rapid adaptation of PPE manufacturing and distribution systems in China to address the growing demand during the pandemic. The significant increase in mask production and the surge in demand for N95 respirators illustrate the urgent need for adequate PPE supplies to ensure the safety and protection of frontline workers and the general public [27].

The WHO has estimated a need for 89 million medical masks each month, 76 million exam gloves, and an international demand for 1.6 million goggles per month [32,33]. Improper disposal practices for biomedical waste and healthcare waste (BMW) can lead to environmental contamination, the destruction of beneficial microbes in septic systems, and the risk of physical injuries from sharp objects. Contaminated soil and groundwater, disrupted septic systems, and potential harm from sharp waste items are key concerns associated with improper BMW disposal. Implementing proper disposal protocols, comprehensive waste management systems, training, and public awareness are vital for mitigating these risks [10].

It should be noted that infectious waste is not limited to hospitals and health centers, as people with minor symptoms or who are asymptomatic also generate contaminated waste, such as disposable masks and gloves [26,33,34]. The World Health Organization (WHO) recommends, due to the pandemic, that solid household and commercial waste generated in homes and businesses in general be collected and disposed of according to usual practices, with no need for any additional treatment. However, the hygiene care practiced by collection professionals and their use of safety equipment should be doubled [29].

Even prior to the pandemic, projections had already indicated a worrisome estimate of approximately 12 billion metric tons of plastic waste accumulating in landfills and the natural environment by the year 2050. Due to this context, the authors point out that efforts must be made to promote recycling, reduce single-use plastics and implement comprehensive waste management systems to address this growing concern [27]. Bown (2019) [35] pointed out that the increased use and consumption of single-use-plastics (SUPs), not only during the COVID-19 pandemic but mostly after this period, will result in an increased demand from plastic suppliers (e.g., in China and the USA).

It is stated that the pandemic has resulted in behavioral changes, leading to greater reliance on disposable plastic utensils, such as cutlery. Unfortunately, existing waste management systems are ill-equipped to effectively handle the influx of plastic waste. As a consequence, this poses a significant danger to both natural ecosystems and human health, prompting considerable deliberation on the issue of medical waste disposal systems [26].

Furthermore, there has been a threat of pollution from plastic waste since the World Health Organization declared the coronavirus infection a pandemic, leading to an increase in household and hospital waste. Benson et al. (2021) reported that plastic-based Personal Protective Equipment (PPE) has been extensively employed as a means of mitigating the risk of exposure to severe acute respiratory syndrome coronavirus 2 (SARS-CoV-2). This includes the widespread utilization of millions of surgical masks, medical gowns, face shields, safety glasses, protective gowns, disinfectant containers, plastic shoes, and gloves, all aimed at minimizing the potential of encountering the virus [36,37].

Plastic items made of non-woven materials (such as certain masks) typically contain polypropylene and polyethylene, which degrade into smaller microplastic particles. Consequently, the utilization of these face masks by non-professionals contributes to a significant environmental issue, amplifying microplastic pollution in marine and freshwater ecosystems [27,34].

Dozens of disposable masks were found on a beach in the Soko Islands in Hong Kong, according to the NGO Oceans Asia. Also, in the Magdalena River, in Columbia, the degradation of non-woven synthetic fabrics was the predominant origin of microplastic microfibers found in samples of water and sediments [26,36]. The Organization for Economic Cooperation and Development (OECD) states that countries must ensure that all cities guarantee the collection of waste, but the waste is not necessarily separated into specific types, and it has proposed the closure of some recycling centers [10].

Modifications were implemented in municipal solid waste (MSW) management services during the COVID-19 pandemic in both developed countries like the United Kingdom, USA, Singapore, and Japan, and in developing countries, including India, Malaysia, Brazil, Indonesia, and Vietnam [13,38,39,40,41]. It is worth noting that a significant portion of MSW generated in the latter group of countries is disposed of in landfills and open dumps due to the lack of incineration facilities [10]. Moreover, developing countries often lack essential infrastructure such as sealed trash bins and plastic bags, leading to the improper disposal of infected or hazardous waste alongside general municipal solid waste [10].

Furthermore, the COVID-19 pandemic can give rise to significant environmental pollution issues due to the production and generation of microplastics (MPs), as highlighted by Fadare and Okoffo (2020) [12]. MPs are categorized into primary and secondary types. Primary MPs are intentionally manufactured to be small, such as microspheres. On the other hand, secondary MPs are derived from larger plastic fragments that have undergone degradation and decomposition over time due to physical, chemical, and biological factors in the environment [42].

The most prevalent types of MPs found in the environment include polyethylene (PE), polyethylene terephthalate (PET), polypropylene (PP), polystyrene (PS), polyvinyl chloride (PVC), nylon-polyamide (PA), and acrylonitrile butadiene styrene (ABS), as well as various copolymers and plastic mixtures [43,44]. When these materials are discarded in the natural environment, they can take hundreds of years to degrade [38,39]. Plastic waste-induced environmental pollution is a growing global problem, with discarded plastic products and plastic debris (MPs) ending up in water bodies and oceans, having detrimental impacts on marine ecosystems.

Therefore, the practices that are currently saving lives may inadvertently contribute to environmental harm in the future, underscoring the significance of proper disposal methods for the aforementioned materials and the implementation of effective waste management and treatment systems to prevent the emergence of new problems stemming from the ongoing pandemic. The extensive production of millions of PPE items made from synthetic materials like PE, ABS, PVC, and others to meet the demands of the COVID-19 pandemic has raised concerns among environmental agencies.

According to a survey conducted by PlasticsEurope (2018) [45], the global production of plastics has increased considerably in the last 60 years, reaching 359 million tons in 2018, with the largest generators being in Asia (51%, with China alone accounting for 30%), in countries belonging to the North American Free Trade Agreement-NAFTA (18%), and in Europe (17%) [45]. Plastics of a wide variety of sizes and origins, including industrial [46], domestic [47], and medical, are present in the environment. Figure 1 shows the main types of PPE used, the materials they are composed of, and their impacts on the environment.

The materials used in PPE, such as NWF, polyethylene, and plastics, are primarily derived from petroleum and are not easily biodegradable. These items are typically used for short durations, and depending on their disposal methods, their accumulation can lead to significant environmental impacts, particularly with respect to the buildup of waste in landfills and the bioaccumulation of microplastics. It is important to address the proper disposal and management of these materials to minimize their negative environmental effects.

Plastic reduction policies and plastic waste management strategies have recently experienced setbacks or temporary delays due to COVID-19 as the prioritization of human health has taken precedence over environmental protection [48]. The monthly use of PPE has reached 129 billion masks, 65 billion gloves, and 1.6 million goggles [49] worldwide, generating a significant increase in plastics on the planet, which are subsequently transformed into MPs. Microplastic contamination in marine environments is serious and has become a global concern due to its wide and growing distribution.

It can be stated that the potential environmental risks associated with microplastics (MPs) include physical abrasion and the obstruction of ingestion pathways in marine organisms [50]. Other hazards arise from the leaching of toxic additives and MP monomers [51], from the absorption of persistent hydrophobic organic pollutants and heavy metals present in MPs [47,48], and from the transport of microorganisms and pathogens associated with MPs [50]. The toxicological risks of microplastics are further amplified via the process of bioaccumulation (transference through the food chain), wherein aquatic organisms at higher trophic levels can be exposed to stronger adverse effects [51,52,53,54,55,56,57].

Human beings are exposed to plastic debris through the consumption of seafood and drinking water and via contact with food and beverage packaging and other materials, such as PPE. The accumulation of MPs in humans presents potential health risks, including cytotoxicity, hypersensitivity, unwanted immune responses, and acute responses, such as hemolysis. In the scientific literature, it is possible to observe experiments conducted to investigate cellular responses upon contact with primary and secondary polypropylene microplastics (PP) of approximately ~20 μm and 25–200 μm, respectively [58].

The results showed that the presence of PP particles in the medium, especially those below 20 μm, were cytotoxic, and that this toxicity was caused by an increase in ROS (reactive oxygen species) and occurred as a function of size and concentration. However, larger PP particles and PP powder particles showed less cytotoxicity. The authors concluded that cells that come into direct contact with PP particles pose a potential health risk via the induction of cytokine production from immune cells rather than direct toxicity to cells. They noted that there are thousands of other types of plastic in various concentrations and size configurations that should be studied [58].

Experiences and successes in Wuhan suggest that improving the emergency management system for medical waste is crucial to mitigating risks to human health. Therefore, the following four steps can be followed: (1) implementing a sophisticated medical data system, (2) improving hospitals’ medical waste storage capacities to handle significant increases during emergencies; (3) developing emergency plans to coordinate disposal resources across the region; and (4) strengthening Wuhan’s emergency response capacity through collaboration and support from other areas in the country [26].

### 3.1. Sustainable Personal Protective Equipment to Mitigate Environmental Impacts

Life cycle assessments (LCAs) study environmental aspects and possible impacts on the environment through the life cycle of a product, that is, from the cradle to the grave, from the acquisition of the raw material and through the production system to its use and final disposal. Through the analysis of environmental impacts, such as climate change, the depletion of fossil fuels, the depletion of water, marine and freshwater ecotoxicity, and marine and freshwater eutrophication, it is possible to measure how much something will harm the environment from the extraction of raw materials until its conception and final disposition [59].

Personal Protective Equipment is important to discuss since its use in the face of the COVID-19 pandemic has become essential. Research regarding the use of life cycle assessments, measured emissions, and the waste generated from locally produced reusable face masks and disposable surgical face masks has also been conducted. The results of the LCAs of both show that the use of a reusable embedded filtration layer (EFL) face mask will generate less waste and will have an impact of at least 30% less among the impact categories considered when compared to the use of single-use surgical mask, indicating it as a popular alternative to the use of reusable masks to mitigate environmental impacts [60].

#### 3.1.1. Biodegradable Materials

Biopolymers are polymers produced from raw materials from renewable sources, such as corn, cassava, cellulose and others, and have shorter life cycles when compared to those of fossil origins, such as polyurethane. Biopolymers are factors of environmental and socioeconomic interest due to the mitigation of the environmental impacts of oil extraction and refining. Studies have identified some technical limitations of biopolymers due to their properties, such as their thermal resistance, mechanical and rheological properties, and the applicability of their properties on an industrial scale [61].

Regarding the development of Personal Protective Equipment with biodegradable materials, the market for biopolymers stands out, including materials derived from plants, biomasses, celluloses, and even microorganisms, many of which stand out for their excellent properties, such as poly (lactic acid) (PLA), which is a kind of aliphatic polyester produced via the fermentation. of sugar, which demonstrates biodegradability, biocompatibility, non-toxicity, a high level of mechanical resistance, and a good cost/benefit ratio. PLA has been widely studied and used for food packaging and tissue engineering applications and can be an attractive field of study for the construction of biodegradable PPE [62]. In particular, PLA is known as a radiation-degradable polymer, and there are records of its complete degradation in a period lasting from six months to one year [63].

Another very widespread polymer like PLA is polybutylene succinate (PBS), which is obtained from the condensation polymerization of succinic acid (AS) and butanediol. It draws attention for its thermal and mechanical properties [64]. Bacterial cellulose, derived from several microorganisms, is also attractive in the manufacture of numerous materials, including for PPE, given its wide applicability and commercial advantages [65]. Associating several polymers to make Personal Protective Equipment is attractive in view of their life cycle analyses, from the extraction of the raw materials to the equipment’s final disposal, helping to mitigate the environmental impacts caused by polymers derived from petrochemicals.

Research on the production of 3D printed safety protective devices, with a focus on the production of respiratory masks in response to the COVID-19 pandemic, has been observed. Topics such as material selection and assessments of mechanical strength and biological safety, as well as analyses of the mechanical and safety characteristics of masks, have been covered. The study concluded that masks 3D printed using home-grade printing equipment have similar levels of performance to industrial-grade masks, and it develops new approaches for the post-processing phases of additive manufacturing, aiming to ensure human safety in the production of personalized medical devices that have been 3D printed [66].

The use of 3D printers can be observed in various studies, such as the case of a project involving a fibrous mask filter made with polybutylene succinate and microfiber and nanofiber mats and coated with chitosan nanowhiskers. It is worth noting that a wheat gluten biopolymer was used as a filtering medium in face masks, and an air-permeable mask was developed using electroshocked licorice roots. A biodegradable mask filter was made via electrospinning and 3D printing polylactic acid which filtered 79% of the air at a particle size of 500–600 nm, which is superior to standard face masks. Polylactic acid was suggested as a suitable material for reusable respirators, and its microstructure was not affected after the efficient disinfection of bacteria, fungi and viruses [67].

### 3.2. Waste Treatment and Management Systems

Some of the largest environmental problems caused by the pandemic are municipal solid waste (MSW) and hazardous biomedical waste. The proportion of non-infectious waste, which is more than 80% of the total amount of health waste generated, needs to be collected and disposed of as municipal waste [29].

The widespread use of protective equipment worldwide in conjunction with the pandemic has led to massive waste management difficulties and improper disposal practices worldwide. The plastic products used are correspondingly pathogenic and should be regarded as hazardous wastes as landfills manage them by promoting the biodegradation of plastics. Plastic waste management was considered a primary environmental concern before the beginning of the COVID-19 pandemic due to increasing concerns about pollution in marine and terrestrial ecosystems [68].

According to the World Health Organization (2020), the use of masks by ordinary citizens quickly became controversial due to the lack of correct handling and disposal and the shortage of this material in healthcare facilities [29]. Guidelines for the disposal of infectious and non-infectious health wastes were established during the outbreak of COVID-19 by the WHO. Procedures for the treatment and disposal of waste at health facilities, as recommended by WHO, involve heat treatment and the use of traditional biocidal agents with proven effectiveness in the destruction of the COVID-19 virus particles [17].

However, the major factors associated with managing MSW outside of health facilities also need to be addressed, such as virus resistance, differences in waste management systems, and the climatic conditions in each affected region [69]. Also, the PPE items generated in large quantities, such as protective masks, which are currently used by the vast majority of the population and are most often incorrectly disposed of as common waste without undergoing any type of treatment, require special attention.

A lack of proper waste management strategies and the uncontrolled combustion of medical plastic waste has accelerated the release of greenhouse gases (GHGs) and other potentially dangerous compounds, such as dioxins, PCBs, furans, and heavy metals, creating significant environmental concerns. The COVID-19 pandemic has placed this issue on the front line of environmental research through the increase in single-use plastics. In addition to increasing the use of Personal Protective Equipment (PPEs), there was in an increase in the plastic packaging of foods and groceries intended for home delivery during the lockdown and home quarantine periods [68].

The integration of waste management in disaster management planning will result in inclusive response measures and guidelines for operating better in the dynamics of a future pandemic, prioritizing the formulation and implementation of homogenous plastics, eco-friendly bio-plastics, and circular technologies while phasing out single-use plastic through taxation. To safely manage biomedical wastes, an automated system of waste storage, collection, treatment, and disposal should be developed using advanced technologies and the Internet of Things [30].

The US Occupational Safety and Health Administration (OSHA) has established predefined safety guidelines for personnel involved in health waste management, recognizing it as an essential service and requiring employees to take appropriate precautions [70]. The European Commission has formulated a document that emphasizes the importance of continuing adequate MSW management services, including separate collection and recycling in accordance with EU law, further specifying the need for proper sorting for the separation of recyclables and biodegradables [71]. However, there is greater concern regarding the handling of medical waste and waste generated in infected homes in less-developed countries, such as India and Malaysia, where little attention is paid to the management of MSW [13].

In Spain, the regulations currently allow infectious waste to be co-incinerated with other waste for use in cement factories. Norway has temporarily authorized landfills for the final disposal of infectious waste, as well as the transport of waste to other disposal sites, due to the increase in the generation of this type of waste [72].

In Brazil, the Brazilian Association of Sanitary Engineering (ABES—a local acronym) has prepared a document advising contaminated patients who are in home isolation to pack their generated waste in double bags up to 2/3 full, tightly close them, and leave them out for conventional collection. Regarding recyclable waste, this is to be stored at home during quarantine for an undefined period, paralyzing the activities of many recycling associations [73].

The Brazilian Association of Public Cleaning and Special Waste Companies (ABRELPE—alocal acronym) has a different position and recommends the continued separation of recyclable waste for those individuals who are not infected by the virus [74]. The two entities agree on the orientation of waste management in environments with a high concentration of people, such as buses, subways, trains, hotels, highway service stations, ports, and airports, among others, where waste must be disposed of as health waste, classified, according to Brazilian Health Regulatory Agency (ANVISA—Local acronym) Resolution 222, as biohazardous waste-Group A1 [75].

The United Nations Environment Program (UNEP) has prepared nine technical sheets with information that can help individuals, companies, and government authorities manage the waste generated during the pandemic, namely:▪Sheet 1—Introduction to COVID-19 waste management;▪Sheet 2—National medical waste capacity assessment;▪Sheet 3—How to choose your waste management technology to treat COVID-19 waste;▪Sheet 4—Policy and legislation linked to COVID-19 pandemic;▪Sheet 5—Links with the circularity of non-hospital waste;▪Sheet 6—Linkages of air quality and COVID-19;▪Sheet 7—Household medical waste management strategies;▪Sheet 8—Disaster and conflict; and Sheet 9—COVID-19, wastewater, and sanitation [76].

The great challenge pointed out by UNEP is the objective of avoiding possible long-term impacts on the environment via the available waste management solutions. To this end, within its short-term recommendations are (i) to manage the increase in waste production, maximizing the use of existing facilities; (ii) ensure that operations respect emission limits and thus avoid secondary health impacts; (iii) in the absence of appropriate technology, consider adopting the 3S methodology (sorting, segregation, and storage—classification, segregation and storage) and install temporary/palliative solutions [76].

Incorrectly managing and disposing of waste during the pandemic can further spread the virus, especially in developing countries, due to poor waste-handling conditions associated with the inappropriate use of Personal Protective Equipment and poor sanitation conditions [69]. Studies on the management of plastic waste generated during the pandemic, within addition to the generation of some ideas about solid waste management, have been conducted [77]. Research on the positive and negative effects of the pandemic on the environment, emphasizing concerns such as the increase in healthcare waste volume and the delay in waste recycling activities, which can have a negative impact on the environment, has been observed [78].

Hospital waste is usually incinerated, turned into energy, or disposed of in landfills [79]. In more developed countries, this waste often goes through a sterilization process before any disposal strategy, preventing the proliferation of diseases. However, large disparities still exist globally, and in developing countries, achieving the correct disposal of hospital waste is problematic, resulting in much of it being dumped in landfills.

While some countries or municipalities are able to properly manage this waste, others are being forced to apply inadequate management strategies, such as direct landfilling or burning [27]. The significant contribution of PPE during the pandemic period constitutes a logistical challenge in relation to the provision of waste management services. Even in countries with significant recycling rates, like India, with 60% [80], it has been noted that inadequate waste disposal procedures and even burning, have increased substantially in some municipalities in an attempt to avoid spreading the virus [81].

On the other hand, countries with larger economies managed to overcome the adversities of COVID-19 in the management of plastic waste. Wuhan was a city that demonstrated efficiency in the disposal of medical waste during this pandemic, even with an increase of almost six times more than normal, reaching almost 247 tons/day. The technology designed by the waste management authority of this city of 11 million people was the distribution of mobile incinerators to safely dispose of the extremely high amount of potentially contaminated PPE waste generated [82].

Wuhan’s medical waste management experience, in the context of the COVID-19 pandemic, can be presented as a valuable example of an emergency response that can inform cities around the world about the formulation of environmental policies that occur simultaneously with pandemic control and other urgent environmental stressors. Despite the lack of capacity to dispose of medical waste in the early stage of the COVID-19 pandemic, Wuhan employed three emergency measures in response to the rapid spread of COVID-19: the use of facilities’ disposal furniture, the expropriation of municipal waste incinerators, and the implementation of external disposal [26].

It can be stated that some lessons have been learned from the pandemic in certain municipalities regarding the management of healthcare waste. The importance of adopting automated systems in which there is no need for human contact in handling this highly contaminating waste, based on Internet of Things technology, which enables the tracking of waste information, has been observed. Additionally, there is a need to maintain larger facilities for medical waste in emergencies like the COVID-19 pandemic [83].

Due to the COVID-19 pandemic, health and safety recommendations have been expanded, prioritizing the treatment of solid waste, especially plastics, through incinerator systems and final disposal in landfills. This has resulted in waste management strategies that lead to an increased use of natural resources for the production of plastics and higher emissions of greenhouse gases and other compounds that pose a risk to the environment [84].

To effect positive changes in the environment, individuals and governments can adopt the following strategies:▪Ensure the regular maintenance of vehicles;▪Implement well-organized public transport systems;▪Improve traffic management systems;▪Reduce emissions of chlorofluorocarbons (CFCs);▪Use eco-friendly products;▪Implement a well-organized and effective waste management system;▪Promote reused and recycled waste materials; and▪Ensure the proper treatment of wastewater before discharging it into the environment [32].

#### Solutions, Approaches, and Technologies for PPE Recycling

The huge amount of PPE can cause harmful impacts to several ecosystems, especially marine wildlife, as PPE debris in marine environments is considered an emerging form of plastic debris and an addition to the existing microplastics crisis. It should be noted here that numerous studies have already documented the impacts of COVID-19 litter on wildlife through entanglement, entrapment, and in-management [85].

A survey carried out on the Indian coast on waste monitoring assessments, conducted at various points along the coast, pointed to ineffective waste management, citing the behavior of the population (social responsibility and public awareness of the disposal of PPE) as one of the fundamental causes of pollution from marine litter. Approximately 60 to 85% of plastic waste in India has been mismanaged, with a tendency for it to enter the environmental matrix, including surface water systems [85].

In recent years, several studies have been published on the ingestion of micro-plastics by marine animals in India. The bioaccumulation of microplastics in mesopelagic and epipelagic fish, the Indian edible oyster, Indian white shrimp, bivalves, and in some commercially important fish and other marine wildlife has been documented very recently [85]. The presence of MPs along terrestrial and marine food chains suggests that humans are exposed through the consumption of contaminated seafood and food products [86].

It should be noted that this problem is not found only in marine environments but on land as well. Abandoned protective masks in the terrestrial system can obstruct urban sewage systems and impact the aeration and percolation of water in agricultural soils. Incorrect disposal of masks can also threaten fauna through entanglement or by being mistaken for food, as in the reported case of a bird entangled in masks and killed in Colombia. The accumulation and translocation of small plastic particles in plant tissues, which can influence plant growth and agricultural productivity, have also been reported [67].

As emerging pollutants, MPs have received global attention due to their wide distribution, high abundance, toxic substance enrichment, and potential threats. Researchers indicate protective masks as new sources of microplastic pollution and proposed the need to take measures to prevent the problem of microplastics derived from PPE [67].

More coordinated engagement is necessary for circular economic approaches, particularly in policies and practices related to the recycling of Personal Protective Equipment (PPE). Various methods such as glycolysis, aminolysis, hydrogenation, hydrolysis, gasification, and pyrolysis are now focused on exploring advanced technologies to convert PPE waste into value-added products. Recent studies demonstrate that the pyrolysis of COVID-19-related PPE waste is the most effective method and an ecologically sound solution with significant potential for application [85,86].

PPE recycling can generate value-added products and mitigate disposal issues while providing energy sources [86]. For example, Eco Eclectic Technologies created “Brick 2.0”, made from recycled PPE face masks (FMs), that can contribute to solving waste disposal issues and provide a value-added product. The composition of the brick comprises 52% crushed EPI materials, 45% waste paper, and 3% binding [85]. Masks have great potential to be applied in the construction of road and rail embankments, sanitary landfills, or recovery constructions.

The reuse of face masks after decontamination is also a strategy to reduce their use and disposal. Efforts were made to decontaminate and reuse FMs to address product shortages and the environmental burden produced. Various methods, such as ultraviolet germicidal irradiation, dry and wet heat treatment, vaporized hydrogen peroxide, and ethanol treatment, have been developed for mask decontamination. Most decontamination methods are tested and proposed for reuse of N95 masks [67,87].

The proper management of used FMs is imperative to decrease the release of MPs into the environment. In addition, the plastics in protective masks can be recycled via mechanical recycling. The direct recycling of masks can be achieved through injection molding or by improving mechanical performance with additives from industrial waste [67]. The direct conversion of face masks into functional materials, particularly carbon-based materials, is also an alternative management strategy. Due to their unique fibrous structure and simple composition, discarded FMs are good raw materials for manufacturing carbon materials for various applications [67].

It is highlighted that certain solutions should receive more attention and more research, including the following:▪Improvements in design, such as reducing the amount of plastic used or replacing it with more eco-friendly alternatives wherever possible;▪In the case of Personal Protective Equipment (PPE), opting for reusable alternatives like cotton masks or treating disposable PPE to enable the reuse of N95 masks that can be decontaminated by steam; and▪Substituting disposable plastics with bio-based solutions (as indicated in Section 3.1.1).

## 4. Sustainable Development Goals and Solid Waste in the Context of the COVID-19 Pandemic

In the current context, although a large part of the world’s population aspires to reach the SDGs by 2030, there was a setback after the COVID-19 pandemic, as investors became more concerned with the rate of return and investment risk than with the environment and SDG indicators [78,79]. The impact of the pandemic on several SDGs is evident [80,81].

Drawing lessons from the current experiences related to COVID-19 highlights the significance of giving greater consideration to the management systems and policies pertaining to climate and environmental matters [32]. Therefore, the 2030 SDG agenda for environmentally sustainable development, which covers sustainability in all forms, can be a useful agenda to form guidelines for sustainable post-pandemic ecological future.

This pandemic, caused by a single virus, has paralyzed nations irrespective of their socio-economic and technological status. The pandemic exposed the inefficacies of contemporary frameworks for sustainability which did not consider global crises of this extent in their design. One such instrument for environmental sustainability, the Sustainable Development Goals (SDGs) framework, has suffered an existential blow due to these new circumstances, exemplifying how the environment truly encompasses every aspect of existence to synergistically benefit humans and nature and cannot be compromised, especially from a policy perspective [88,89,90,91].

In this section, the relationship between COVID-19 and the SDGs is discussed; more specifically, the waste generated from the PPE used in hospital environments is addressed. In the article “COVID-19 and the UN Sustainable Development Goals: Threat or Opportunity for Solidarity?”, the importance of the impacts of the pandemic caused by COVID-19 with respect to the SDGs 1, 2, 3, 4, 5, 8, 10, and 16 was discussed. However, in this study, the focus was on SDGs 8 and 12 as objectives directly achieved due to the protection of health professionals and the generation of solid waste, respectively. Subsequently, SDGs 6, 11, 14, and 15 will be addressed and considered in this study as indirectly achieved objectives [92].

In the case of SDG 8, in the context of the study, the effect of COVID-19 in promoting safer work environments for all workers, including health professionals, is added here. Goal 8.8, for example, is aimed at safe and protected work environments for all workers, including healthcare workers, through the use of Personal Protective Equipment (i.e., masks, gloves, glasses, lab coats, etc.).

The solutions must be directed toward both the protection of the health professionals involved in the pandemic and to the proper management of hospital solid waste. Surgical masks should not be used for more than a few hours and should be properly discarded to avoid cross-contamination since with the incorrect disposal of PPE, the virus can spread quickly in various public places and in the environment. Therefore, it is observed that at the same time that health professionals need protection, accumulations of solid hospital waste are multiplying [16,27].

In this way, COVID-19 also impacted the SDGs by increasing the generation of waste, mainly from hospitals, such as masks, glasses, white coats, and other types of PPE. However, due to the rapid progression of the COVID-19 pandemic, the preventive measures implemented to control and mitigate its high transmission rate demanded a sudden increase in the demand and consumption of plastic products by the general public, health professionals, and service providers [27].

In this sense, SDG 12 aims to ensure sustainable standards of production and consumption, and goal 12.5 aims to substantially reduce the generation of waste through prevention, reduction, recycling, and reuse by 2030 (UNDP, 2015). The consequences of COVID-19 have seriously disrupted waste management policies, especially on plastic reduction at the regional and national levels [5,26].

It is also of great importance to highlight SDGs 6, 11, 14, and 15, which are indirectly related to the current pandemic context. The objective of SDG 6, goal 6.2, is to achieve access to adequate and equitable sanitation and hygiene for all. During this global health crisis, many people in developing countries, like Brazil, do not have access to clean water and basic sanitation. Furthermore, it is necessary to contemplate SDG 11, which deals with sustainable cities and communities.

Goal 1.1 of SDG 11 stands out, aiming to guarantee access for all to safe, adequate and affordable housing and to basic services and to urbanize the favelas by 2030; by 2030, goal 11.5 aims to significantly reduce the number of deaths and the number of people affected by disasters and to substantially decrease the direct economic losses caused by them in relation to the global gross domestic product, including water-related disasters, with a focus on protecting the poor and vulnerable people; and goal 11.6 aims to reduce the negative environmental impact per capita of cities, including paying special attention to air quality, municipal waste management, and others, by 2030.

The targets mentioned above cite issues that were very evident in this pandemic, especially in developing countries; after all, many locations do not have basic services such as sanitation and running water, and millions of people are unprotected because they do not have adequate housing, among other facts that impair the effectiveness of the 2030 Agenda goals. It can be asserted that these impacts are already negative for wealthy countries and will likely be felt even more strongly in developing nations, which lack the capacity or resources to address the numerous economic and social challenges imposed by the disease. Ultimately, the COVID-19 pandemic reveals the urgent need for action in areas such as security, employment, social and public health, environment, among others [92].

Consequently, the indirect negative impacts in relation to compliance with SDGs 14 and 15, resulting from the inadequate disposal of solid waste during the pandemic, are also highlighted. As mentioned in Section 4, the dozens of disposable masks were found on a beach on the island of Soko in Hong Kong and in the Magdalena River in Columbia, illustrating the risks to the conservation and sustainable use of the oceans, seas, and marine resources (SDG 14) and the protection, recovery, and promotion of the sustainable use of terrestrial and freshwater ecosystems (SDG 15).

Figure 2 was adapted to the context of this study, expanding it to include other important impacts of COVID-19 in SDGs 6, 8, 11, 12, 14, and 15 related to the management and treatment of PPE waste [92].

The importance of prioritizing environmental goals still applies in a post-pandemic scenario. Few of the goal targets proved to be especially significant from a pandemic context; lockdowns helped achieve and/or prevent future environmental disasters [30].

Apparently, the COVID-19 outbreak has brought several positive and negative effects on the environment globally. During this outbreak, the GHG emissions, pollutants in the water, noise pollution, et cetera, suddenly decreased due to travel restrictions and the closure of industries and companies. On the other hand, the use of plastic increased in the food and grocery home delivery service to maintain social distancing, hygiene, and cleanliness to reduce the spread of the COVID-19 virus [68].

This COVID-19 pandemic seems to be preserving the UN sustainable development goals (SDGs) 2030 (namely 3, 6, 11, 12, 14, and 15) by reducing pollutants in the air and water [27,68]. However, the increasing use of SUPs, PPE, medical waste, and household waste has directly violated the UN-SDGs (namely, 3.3, 12.3, 12.4, and 12.5).

Finally, after analyzing and studying the literature, Table 1 summarizes the possible relationships between the Sustainable Development Goals and the COVID-19 pandemic. This table was intended to detect issues raised during the pandemic period and thus, this article suggests the continuation of future research and possible solutions for meeting the goals of the SDG.

## 5. Trends, Future Prospects, and Conclusions

The COVID-19 pandemic has brought to light the dependence on plastic disposables and the fragility of solid waste management systems. Among these disposables, notable items include those used as Personal Protective Equipment (PPE), the main line of defense of health professionals which prevents them from becoming contaminated and spreading the virus among patients. PPE must be changed several times a day as it can carry the COVID-19 virus. Therefore, hospital waste multiplied in the face of the pandemic, raising questions about the management of this hazardous material.

Studying the impact of this waste on the world is now an issue of extreme importance, with tons of PPE being produced and discarded daily. In addition to research on better ways to manage hospital waste, investments aimed at producing PPE with biodegradable materials have never been more important in order to achieve a more sustainable life cycle compared to the use of petrochemical components.

In addition to the attention provided to the safety of workers in the handling of PPE waste, especially at the present time, the devices adopted through the application of policies such as the shared responsibility for the life cycle of these products, reverse logistics, sectoral agreements, economic instruments, goals for reuse, recycling and the final disposal of these residues, contributes to minimizing the environmental impacts of PPE waste on the environment, as well as to reducing the use of natural resources.

Furthermore, it should be emphasized that in view of the COVID-19 pandemic, it is essential to reinforce the search for concrete actions and strategies at the federal to the institutional levels, in an articulate manner and among all sectors, for the implementation of guidelines aimed at the improvement of solid waste management practices. In particular, this should address the huge increase in PPE waste, which can contribute to the generation of micro- and nanoplastics in the environment, with adverse impacts on ecosystems.

It was noted that the Wuhan medical waste management experience, in the context of the COVID-19 pandemic, can be presented as a valuable example of an emergency response that can inform cities around the world about the formulation of environmental policies that occur simultaneously with pandemic control and other urgent environmental stressors.

It is of great concern that the pandemic presents a concrete threat to the commitment made by nations regarding the achievement of the UN sustainable development goals (SDGs), especially with respect to the environment, health, and well-being, notably the much-needed reduction in the generation of waste. This study provides an in-depth theoretical insight into the impacts of the use, in large volumes, of PPE in hospital environments, which is necessary for the direct protection of workers and the indirect protection of patients but is generating a serious problem in the form of waste. It also demonstrates the possibility of using biodegradable Personal Protective Equipment to mitigate environmental impacts.

The discussions presented in this article, based on the extensive literature, highlight the adverse effects of PPE due to the materials from which it is produced and its intensive use, with serious consequences in relation to the reach of the UN SDGs 6, 8, 11, 12, 14, and 15. The importance of sanitation and access to water for the hygiene of people, the conditions of protection and safety for health professionals, the influence of management in sustainable cities, responsible production and consumption, waste management, and even impacts on soils and water resources were addressed in this article.

Therefore, it is suggested to encourage new research relating to the management of solid waste and treatments during the pandemic, whether hospital or domestic waste, as well as the importance and need to address topics such as new textiles and smart textiles for the manufacturing of PPE.

## Figures and Tables

**Figure 1 polymers-15-03151-f001:**
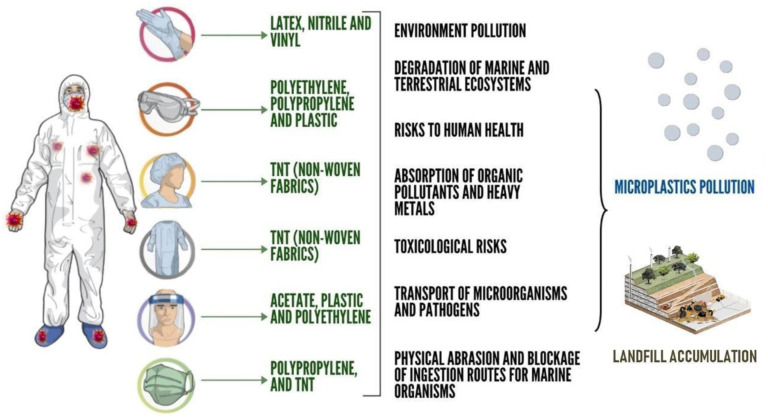
PPE and its impact on the environment in 2020. Source: Authors, 2022.

**Figure 2 polymers-15-03151-f002:**
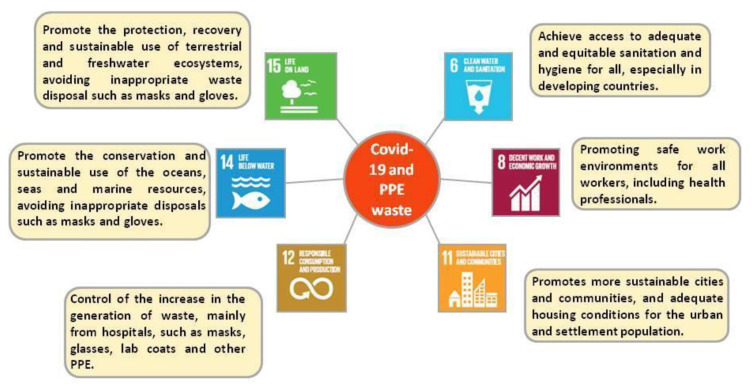
Impacts of COVID-19 on the SDGs. Source: adapted from Leal Filho (2020).

**Table 1 polymers-15-03151-t001:** Relationships between the Sustainable Development Goals and the COVID-19 pandemic.

SDG	Target	Relation to the COVID Pandemic
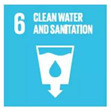	6.2	Many countries do not have access to basic sanitation and adequate hygiene, contributing to the dissemination of the COVID-19 pandemic.
	6.3	To improve water quality, care is needed such as reducing pollution, eliminating waste and minimizing the release of chemicals and hazardous materials.
	6.6	For the protection of aquatic ecosystems, good waste management is necessary. The COVID-19 pandemic showed a significant increase in the inappropriate disposal of PPE, highlighting that many PPE items were located in water resources.
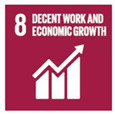	8.8	The COVID-19 pandemic affected the lives of many workers, especially health professionals. Some health professionals were left vulnerable without adequate protection, especially in developing countries.
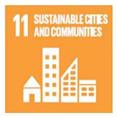	11.1	The COVID-19 pandemic showed the vulnerability of thousands of people without basic housing conditions; in this way, it aggravated the spread of the virus.
	11.5	The COVID-19 pandemic highlighted vulnerability and social, economic, and environmental unpreparedness at a global level. Protection policies against this type of disaster, especially for people in situations of vulnerability and poverty, must be urgently rethought.
	11.6	During the lockdown, a decrease in pollution was observed in some locations. In this way, rethinking post-pandemic social behavior is inevitable.
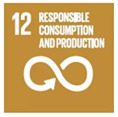	12.4	Through the COVID-19 pandemic, the importance of achieving the environmentally correct management of chemical products and all waste throughout their life cycle was observed. This includes biomedical waste.
	12.5	Through the COVID-19 pandemic, the importance of substantially reducing the generation of waste through prevention, reduction, recycling, and reuse was observed. In addition to thinking about the proper disposal of waste such as PPE and household waste, it is necessary to explore environmentally correct materials, such as upcycling techniques and biodegradable products.
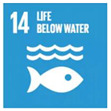	14.1	The COVID-19 highlighted again the connection between the protection of aquatic ecosystems and correct waste management and the need to significantly prevent and reduce marine pollution of all kinds, in particular from land-based activities, including marine debris and nutrient pollution.
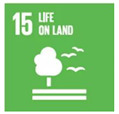	15.2	In addition to the concern regarding terrestrial pollution via inappropriate waste disposal, concerns regarding the conservation of natural habitats, wild animals, and deforestation, among others, were evident. In this way, the proliferation of new viruses is also avoided.
	15.5	

Source: Authors, 2022.

## Data Availability

Not applicable.

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
