# Peer review of "The Impacts of Plastic Waste from Personal Protective Equipment Used during the COVID-19 Pandemic"

_polymers, 2023, doi:10.3390/polym15153151_

Round 1

Reviewer 1 Report (Previous Reviewer 3)

Review of polymers-2432370

This is a resubmitted version of polymers-2259578. The authors have made a substantial improvement to this new version.

Some comments that can be addressed during the post-production process:

  1. The journal “Sustainability” consists of one word only, and therefore must not be abbreviated. Please write it as “Sustainability” instead of “Sustain.”, for references 38, 77, 92.
  2. References 74 and 75: Please do not write in all uppercase letters.
  3. Line 379: What is PCV? Do you mean PVC? (polyvinyl chloride)?

Author Response

Reviewer 2 Report (New Reviewer)

The revision based on the previous reviewer's comments have been adequately addressed. The authors have adequately described the problems associated with disposable PPE's and its effects to the environment. 

There needs to be some moderate editing and proof-reading of the document.

Try not to use the word "Thus" to begin a paragraph, or Thus, since.. 
Pay attention to sentence structure.

Author Response

This manuscript is a resubmission of an earlier submission. The following is a list of the peer review reports and author responses from that submission.

Round 1

Reviewer 1 Report

First of all, thank you for your willingness to share your work “Impacts of the plastics from waste personal protective equipment in the COVID-19 pandemic” with the scientific community.

This review presents an overview of the increased use and waste of personal protective equipment during the COVID-19 pandemic. A review is always a labour-intens task. The general idea is interesting, but the way it is presented is poor. Many paragraphs are a copy-paste from other articles. For example:

Lines 207-210 are a copy-paste of article 30, more specifically, the second paragraph of point 3.2.2.

Lines 211-218 are a copy-paste of part of the abstract of article 31.

Lines 219-225 are a copy-paste of the first paragraph of point 2.2.1. of article 23.

Lines 226-229 are a copy-paste of the second paragraph of the introduction section of article 13.

Lines 229-231 are a copy-paste of the first sentences of page 4 of article 27.

Lines 234-237 are a copy-paste of the second sentence of page 3 of article 10.

Lines 245-247 are a copy-paste of line 12 of the introduction section of article 27.

Lines 250-254 are a copy-paste of sentences from the third paragraph of the introduction section of article 26.

Lines 257-261 are a copy-paste of lines 4-7 of the abstract of article 36.

That means a whole page is a copy-paste of other articles, with the aggravating factor that the original source is not cited.

Author Response

author replied,please check the attachment.

Reviewer 2 Report

The research paper is very interesting. From the environmental perspective, the authors have given an excellent explanation of the impact of plastic from PPE. This paper has been written without guidelines that should be followed to write a review paper. 

Author Response

author replied, please check the attachment.

Reviewer 3 Report

Review of polymers-2259578

  1. Section 1, especially line 274-281: Please add these references about COVID-19 waste management in developing countries:

·       COVID-19 waste management in Malaysia: Waste Management and Research 39 (2021) 18-26 https://doi.org/10.1177/0734242X20959701

·       COVID-19 waste management in Indonesia: Sustainability 14(5) (2022) 2556 https://doi.org/10.3390/su14052556

·       COVID-19 waste management in Bangladesh: Case Studies in Chemical and Environmental Engineering 5 (2022) 100177 https://doi.org/10.1016/j.cscee.2021.100177

·       COVID-19 waste management in India: Environmental Science and Pollution Research 28 (2021) 52702-52723 https://doi.org/10.1007/s11356-021-15028-5

  1. Line 623-629: Please write this paragraph in English.

  1. Please add complete list of abbreviations. Moreover, some abbreviations are mixed up, such as “microplastics” word that is sometimes written as “MPs”, but also “PMs”. Sustainable Development Goals as “SDGs”, but also as “ODS”. Face masks must be abbreviated as “FMs”, not “MFs”, etc.

  1. Line 422-561, or Section 3.2: Please adhere to the MDPI template, use spacing of 0.95 pt.

  1. Line 46: SARS-COV, not SARS-VOC
  2. Line 489-497: Please write this paragraph with bullet points
  3. Line 537: Please add a dot between “stressors” and “Despite”, in order to indicate that these are two separated sentences.
  4. Line 555-561: Please write this paragraph with bullet points
  5. Line 581: Please change “PMs” to be “MPs”, for microplastics.
  6. Line 591: Please change “PMs” to be “MPs”, for microplastics.
  7. Line 604: Please change “MFs” to be “FMs”, for face masks.
  8. Line 610: Please change “MFs” to be “FMs”, for face masks.
  9. Line 615: Please change “PMs” to be “MPs”, for microplastics.
  10. Line 615: Please change “MFs” to be “FMs”, for face masks.
  11. Line 621: Please change “MFs” to be “FMs”, for face masks.
  12. Line 669: Please delete “Natural”. The sentence “in the environmental” is enough.
  13. Line 672: Please change “ODS 12” to be “SDG 12”
  14. Table 1, column 3, line 1: Please write COVID-19 with all uppercase letters.
  15. Line 791-792: ….or domestic waste, as well as, the importance… --> This is actually one sentence, not two separated sentences.
  16. Line 795-798: Please adhere to the MDPI template when writing the Author Contribution, please use initials instead of full name.
  17. Line 798: What do you mean with the solitary “P” there?

  1. References: Please write complete list of authors, do not use “et al.” Please revise Reference 1, 2, 7, 10, 23, 27, 30, 38, 39, 43, 49, 56, 61, 62, 71, 76, 77, 84
  2. Reference 37: Delete “no. xxxx”
  3. Reference 71: What do you mean with “B. Pr” ??? This is such as short unusual last name. Please correct it. It seems that this is a name of an institution. Please write the name of the institution completely in this section.
  4. Reference 88: The journal “Sustainability” consists of one word only, and therefore must not be abbreviated. Please write it as “Sustainability” instead of “Sustain.”

Author Response

(The authors gave the same response as above.)
